# Creating Organisational Working Conditions Where Nurses Can Thrive: An International Action Research Study

**DOI:** 10.3390/nursrep15030095

**Published:** 2025-03-12

**Authors:** Stephen Jacobs, Willoughby Moloney, Daniel Terry, Peter A. Lewis, Annie Topping, Marcela González-Agüero, Stephen Cavanagh

**Affiliations:** 1School of Nursing, The University of Auckland, Auckland 1010, New Zealand; 2Betty Irene Moore School of Nursing, University of California Davis, Davis, CA 95616, USA; wmoloney@ucdavis.edu; 3School of Nursing and Midwifery, University of South Queensland, Ipswich, QLD 4305, Australia; daniel.terry@unisq.edu.au; 4Nursing and Midwifery Workforce and Education Research Group, Institute of Health and Wellbeing, Federation University Australia, Mount Helen, VIC 3350, Australia; 5School of Nursing, Midwifery and Social Work, The University of Queensland, St Lucia, QLD 4072, Australia; peter.lewis@uq.edu.au; 6Department of Nursing and Midwifery, College of Medicine and Health, University of Birmingham, Birmingham B15 2TT, UK; a.e.topping@bham.ac.uk; 7University Hospitals Birmingham NHS Foundation Trust, Birmingham B15 2GW, UK; 8Escuela de Enfermería, Pontificia Universidad Católica de Chile, Santiago 8320165, Chile; mmgonzal@uc.cl; 9Betty Irene Moore School of Nursing, UCDavis, Davis, CA 95616, USA; sjcavanagh@ucdavis.edu

**Keywords:** burnout, co-design, co-production, early-career nurses, nurses, thriving at work, retention, workforce development

## Abstract

**Background**: Attracting and retaining sufficient numbers of nurses is an international challenge. The group most difficult to retain are newly qualified nurses within their first five years of practice or earlier. A recent US study reported that approximately 25 percent of nurses leave within the first year of graduation. Health organisations play a crucial role in providing workplace cultures where nurses feel empowered and can thrive. Research needs to focus on improving organisational culture, yet most approaches to supporting and retaining nurses have used top–down, management-designed interventions. This article describes a collaborative international programme of research. **Methods**: This innovative international theory-driven multi-site action research programme adopts a longitudinal co-design approach based on principles of appreciative inquiry to develop and implement organisational support for newly qualified nurses. It integrates the Institute for Health Improvement (IHI) Framework for Improving Joy at Work and the Thriving at Work model, both focused on improving the well-being of the healthcare workforce and health service outcomes. Each year, a new group of nurses during their first-year orientation is invited to participate. Over five years, each cohort will then participate in an annual survey, focus groups, and co-design meetings with nurse leaders/managers, generating new solutions developed through open dialogue for subsequent testing driven by these key stakeholders. **Expected outcomes**: This research will generate a new co-design management model to improve systems of support that may assist nurse retention and thriving that can be shared with other nursing organisations. It will provide an understanding of the effectiveness of current support for nurses by their employers from the perspective of those nurses whilst providing evidence about what extra support nurses would like from their employers. **Conclusions**: This international research programme gives agency to nurses and organisational nurse leaders/managers to co-design interventions for building positive work environments where early-career nurses can thrive. This programme will capture what works, where, how, and with whom, ultimately benefiting both individual nurses and the overall effectiveness and sustainability of healthcare systems.

## 1. Introduction

### 1.1. Contextualisation

Nurses, irrespective of their employment setting, play a pivotal role in improving the health of the communities they work with. Nurses constitute almost 60% of all healthcare professionals internationally, and there are an estimated 29 million nurses currently practicing [1]. Despite the evidence that nurses have a strong influence on health outcomes, worldwide nursing shortages are predicted to increase [2]. Recruiting and retaining a sufficient number of nurses is an international challenge [1]. The group that employers find is most difficult to both attract and retain has consistently been those within the first five years of practice, referred to as early-career nurses (ECNs) [3,4,5]. A recent study of over 700,000 nurses from across the US found that approximately 25 percent of nurses leave within the first year of graduating [6]. In Canada, 18% to 30% of ECNs choose to leave the nursing profession in their first year, and 37% to 57% leave in their second year [7]. The COVID-19 pandemic brought unique stressors to the job and further exacerbated the challenges of ECN retention [8].

Key factors identified that contribute to nurse attrition and difficulty recruiting applicants are stress, inadequate pay, moral distress, issues with work–life balance, and the toxic leadership styles of some nurse leaders/managers [9]. Burnout has also been identified as one of the major issues, with 31.5% nurses reporting it as the primary reason for leaving employment in 2017 [9]. Beyond the negative impact on individual nurses, burnout is associated with poor quality of patient care, decreased patient satisfaction, and lower organisational commitment and productivity [10,11].

Resilience is a characteristic of a vibrant nursing workforce that has been a major focus of various interventions [12] but that fails to place attention or responsibility on organisational culture. Adverse job characteristics are key factors in burnout and include high workloads, long shifts, poor renumeration, low staff-to-patient ratios, and low control [13]. Importantly, attending to these factors moves the focus away from the resilience of individual nurses towards organisational values that create supportive environments [14]. Given organisations can empower their workforce by providing working environments that enable employees to find meaning and purpose by exercising personal agency to achieve work goals [15], research should focus on what organisations can do to improve working conditions and cultures for nurses and specifically ECNs.

### 1.2. Literature Review and Theoretical Framework

Given the complex nature of health systems, it is evident that management and leadership do not have all the answers to many issues, including workforce management [16]. This is particularly relevant as new generations of ECNs join the system, with different perspectives and attitudes about their work expectations [11]. New data show remarkably different attitudes to work between Millennials (Generation Y) and Gen Z, and as a consequence, managers may have to adapt their management styles to retain younger workers [17]. Given the complexity of the healthcare setting, it is imperative to find novel solutions; the first crucial step in change is to engage with ECNs to ask them for their views, co-operation, and support to develop interventions that might encourage them to stay with employers.

This programme of research theoretically integrates two models that focus on improving the well-being of the healthcare workforce and health service outcomes: the IHI Framework for Improving Joy in Work [18] and the Thriving at Work model [18]. Both models use a positive organisational scholarship approach to identify what is working well and what needs to be improved to best support employees, in this case, ECNs. This focus on listening to the voice of the ECNs and identifying what needs to be improved provides a framework that enables all stakeholders involved to move away from the negativity that often bedevils health workforce discussions. The impact of a negative work environment can be profound. Nurses tasked with providing care under often stressful conditions find their work environment doubly strained when negative cultures prevail. The research approach in this study is being used intentionally to promote a positive environment in which both ECNs and managers can feel supported and heard [19].

The Joy in Work model proposes that having healthy employees is not just a matter of avoiding burnout or an issue of an individual’s well-being; “joy in work” is generated or undermined by the system [18]. Managers within the system, therefore, need to create organisations in which everyone has the capacity to experience joy in their work. The model is particularly appropriate for ECNs as it emphasises creating a joyful, engaged workforce by addressing burnout and other adverse features of unhealthy cultures whilst promoting well-being. This model includes nine critical components that ensure a supportive work environment, which is crucial for ECNs who are at the beginning of their careers and may be more vulnerable to stress and burnout. By focusing on what matters to nurses while co-creating strategies to improve their work environment, this framework aligns well with the needs of ECNs.

On the other hand, the Thriving at Work model provides a comprehensive approach to building mentally healthy workplaces. It focuses on organisational culture, the working environment, and support for individuals, which are essential for fostering resilience and well-being among ECNs. It is also designed to be adaptable to various international contexts, making it applicable irrespective of the healthcare settings. Both models emphasise creating supportive, positive work environments, thus making them highly relevant for addressing the challenges encountered by ECNs and improving workforce retention globally.

As such, the overall framework presents four steps leaders can take to build an organisational environment that supports employees to enjoy work: (1) asking the staff, “what matters to you?”; (2) identifying unique impediments to joy in work in the local context; (3) committing to a systems approach to making joy in work a shared responsibility at all levels of the organisation; and (4) using improvement science to test approaches that seek to improve joy in work in the organisation [20].

The Joy in Work model has been applied with health service workforce [20] and nurses [21,22]; however, there is little empirical evidence specifically of its application with ECNs [22]. The available evidence identifies what matters to ECNs rather than applying the Joy in Work model to improve how ECNS could be supported. As such, this research programme focuses on implementing the four-step Joy in Work model to achieve improvement in organisational supports for ECNs.

The Socially Embedded Thriving at Work model [18] provides the theoretical systems approach required for Step 3 of the Joy in Work model. It focuses on identifying the personal and organisational factors that influence the ability of staff to be energised and facilitate thriving, thereby reducing their intention to leave the organisation or profession [23,24]. Thriving is indicated by the joint experience of (1) vitality, the sense that one is energised, feels alive at work, and has a zest and enthusiasm for work, and (2) learning and growing through acquiring new knowledge and skills [18]. Experts across multiple industries have shown thriving at work to be critically important for creating sustainable organisational performance [25]. Paying attention to thriving at work is an important means by which managers and their organisations can improve both employee health and unit performance [26]. The Thriving at Work model pays particular attention to two issues: (1) the fact that an organisation has a responsibility to provide work environments that support the workforce; and (2) the identification of individual health and developmental outcomes that organisations can use to determine areas where workforce support is needed or can best be targeted. This includes enhancing engagement and satisfaction [26], as well as improving employees’ career self-management through upskilling and feedback-seeking [27,28].

While thriving at work is the psychological state in which individuals experience both a sense of vitality and learning at work [19,27] it is not automatically achieved through the removal of or reduction in stressors. Instead, thriving at work requires an increase in favourable individual, relational, and contextual characteristics [25]. Therefore, the Thriving at Work model suggests that both organisations and individuals are responsible for nurses’ ability to thrive [27] and that the resources available to nurses within organisations are a determining factor in enabling them to thrive [29,30].

### 1.3. Relevance to the Research Programme

A recent meta-analysis of the Thriving at Work model identified a network of antecedents and outcomes associated with the model [25]. These antecedents of thriving at work were further separated into individual characteristics and relational resources. The outcomes included health, attitude, and performance. This conceptual nomological network of assumed antecedents and outcomes of thriving at work has been narrowed for this research, with antecedents excluded if there was no reliable scale or metric to measure the value (Figure 1).

The revised model informed the development of this programme of research and the instruments used to measure constructs. The current literature and previous research were interrogated to identify what the most useful aspects from each of the six components were to measure. Our research programme begins by identifying what factors nurses indicate are contributing to their intentions to leave their organisation or the profession, but innovatively, it then brings nurse leaders and nurses together to develop management support interventions annually in a process of ongoing mutual engagement and collaborative working to bring about a positive change. Within this context, the research seeks to utilise an Appreciative Inquiry framework, emphasising what is working well within the system and the organisation [31,32] and, subsequently, what can be improved. This avoids seeking to identify problems and any focus on what is wrong, thereby reducing any negativity bias and instead adopting a possibility-focused, strength-based, and affirmative approach which aligns with our overarching research aim of devising a supportive mechanism for ECNs [33]. As such, the approach is multifaceted, as it not only addresses any immediate concerns but also fosters a culture of continuous improvement and collaboration and culture change. By engaging both ECNs and nurse leaders in the development of these interventions, our programme of research seeks to create sustainable, adaptive support systems that can evolve in response to the challenges and needs experienced by ECNs and health services. This collaborative approach is anticipated to enhance job satisfaction, reduce turnover rates, and ultimately improve patient care outcomes. Further, data from multiple international sites provide the opportunity to compare results across countries and enhance learning about the most effective tested interventions.

### 1.4. Objectives and Aims

In line with the methodological approach, six universities in five countries are collaborating on a research programme that will continue for five years. They are the University of Auckland, New Zealand, University of California Davis, USA, University of Queensland and University of South Queensland, Australia, University of Birmingham, England, and Pontificia Universidad Católica de Chile. The following overarching research questions guide the objectives of this study:What do ECNs identify as what matters most to them at work?When ECNs and managers are facilitated to meet in an open dialogue, what solutions emerge?Have any solutions been trialled, and if so, did they produce positive results?Do the instruments or data capture methods proposed measure what supports ECNs and what could be improved?Are there cultural or healthcare systems differences that require different solutions in different countries?

The aims of this study are as follows:To provide an understanding of the usefulness of current support mechanisms used with ECNs by their employers from the perspective of ECNs.To generate imaginative suggestions about what workplace support ECNs need to thrive at work.To generate a theory-driven co-design model to improve systems of management support that may assist ECN retention and thriving and that can be shared with other nursing organisations.

## 2. The Research Approach

### 2.1. Action Research

Action research is an approach that involves collaboration to develop a process through knowledge building and social change [34]. Only through action is legitimate understanding possible; a major purpose is to effect desired change as a path to generating knowledge and empowering stakeholders. Action research develops shared learning platforms, alongside people with a stake in transforming structural forces that inhibit thriving [35]. This approach supports the aim of this research programme.

### 2.2. Co-Creation

Co-creation is central to this research programme. Co-creation has been used extensively within healthcare to involve patients in the care planning process, design of services, and interprofessional collaboration and to build a relationship between the healthcare professional and the patient to improve patient satisfaction and well-being [36]. There is a small but growing interest in co-design approaches applied to the nursing workforce [37,38]. We will bring ECNs together with their managers to co-design recommendations for organisational change and then co-produce those changes.

Co-design promotes the creation of shared value by engaging diverse stakeholders in the process of understanding complex problems and designing and evaluating contextually relevant solutions [36]. Involving all key stakeholders is essential to ensure the optimal design, implementation, and evaluation of resulting initiatives [39]. Unlike traditional interventions, which are typically designed by experts with limited user input, co-design emphasises collaboration, flexibility, and contextual relevance. This approach fosters a sense of ownership among participants, leading to more acceptable, usable, and sustainable solutions. Co-design is iterative, allowing for continuous feedback and adjustments, making it more adaptable to specific contexts. By prioritising the lived experiences of end-users, co-design creates interventions that are more effective and better aligned with the target population’s needs and preferences. This does mean that solutions are often local-focused and not obviously transferable to other areas; however, if they are theory-driven, they may share components amenable for adoption beyond the immediate setting. Conversely, co-creation has a focus on the process rather than the result [40,41] and following the Vargas model enables researchers to facilitate contact between stakeholders in which they follow strategic co-design and co-production guidelines [40] (Figure 2).

### 2.3. Co-Production

In this programme, ECNs are informed of the process and that at each stage they are being asked to (re)consent to participate in surveys, focus groups, and co-creation meetings. Alongside ECN engagement, nurse leaders meet with the researchers regularly as an operations group, so they are fully involved and understand that they are committing to listen to the ECNs and to work with them to identify what can be improved. ECNs may also be invited to have a representative on this group. The co-production aspect requires nurse leaders/managers committing to work with ECNs to develop interventions. All parties are made aware and understand the repetition of the process annually is an element of the programme, so ECNs are empowered to provide input and feedback each year regarding differences and improvements from the previous year while voicing what changes made little difference. Nurse leaders are also aware they will be required to report back to ECNs annually regarding the progress of initiatives that have been developed and implemented in response to the information they have received from the ECNs.

At the core of this process is enabling nurse leaders and ECNs to discuss with each other what improvements can be made and what they can work on together using a systematic approach that provides a combination of asking ECNs what matters to them aligned with the co-design and co-production elements. This approach will lead to a greater sense of trust between ECNs and their nurse leaders and a greater appreciation of being heard and, therefore, a sense of belonging [40]. Consequently, the hypothesis is that this increased sense of belonging and trust through the approach will lead to ECNs being more likely to thrive at work and remain in the profession.

### 2.4. The Research Process

Each principal investigator will recruit a study site such as a local hospital or health service and establish an operations group, so that the research operates as a partnership. Each site may have slightly different approaches to the annual cycle to account for local conditions; for example, each study site may have different patterns for recruiting ECNs. Each study site may identify different areas (specialities, wards, units, departments, services, etc.) where ECNs and managers are able to work together to effect change. The study design will encourage study sites to generate ideas locally and local solutions for adoption in each country. The rationale for taking a collaborative international approach is to provide support and to facilitate sharing of ideas, learning, and results. Interventions and anonymised outcome data will be shared across countries to compare and contrast similarities and differences.

There are three main steps in the operation of this programme of research.

#### 2.4.1. ECN Survey

After approximately three months of practice and then at the end of each year of their employment for five years, ECNs will be invited to complete a survey eliciting demographic information including birth year, gender, race/ethnicity, highest level of education, and email for future contact. The survey contains questions on thriving at work, burnout, leadership, organisational support, collegial support, integrity at work, intention to leave, and quality of care (Appendix A). The survey instrument also includes open-ended questions inviting participants to describe what is going well for them in their workplace and what they would like to see improved using free text.

#### 2.4.2. Focus Groups

Participants will be asked in the survey whether they want to participate in focus groups. Those who volunteer will be asked to join a focus group with other ECNs. The primary aim of this focus group is to provide a more in-depth understanding of ECNs’ expectations of what management can do to facilitate their thriving at work. This will add to what was learned from the open-ended questions in the survey. To ensure ECNs feel safe when providing feedback, every effort will be made to protect anonymity and confidentiality. Participants will be informed about these protective measures to reassure them their insights or feedback will anonymous and not be traced back to them.

#### 2.4.3. Co-Design Meetings

Data from surveys, focus groups, and co-design meetings once analysed will be integrated to generate actionable outcomes through a structured and iterative process. Initially, quantitative data from surveys will be statistically analysed using IBM SPSS Statistics (Version 27) to identify trends and patterns, while qualitative data from open-ended survey responses and focus groups will be thematically analysed using NVivo© to uncover common themes and insights. These findings will be synthesised in preparation for the co-design meetings to create a comprehensive understanding of the issues faced by ECNs. These integrated data will then be presented to both ECNs and nurse leaders during co-design meetings, ensuring a feedback loop that validates the data and informs decision-making.

In the co-design meetings, ECNs and nurse leaders will collaboratively prioritise key issues and develop tailored interventions. These interventions will be implemented with a detailed plan, including timelines, responsibilities, and evaluation metrics. For example, the success of any changes or interventions will be evaluated using a combination of retention rates, changes in survey scores, and qualitative feedback from focus groups and interviews. These measures will provide a comprehensive assessment of the interventions. As such, continuous monitoring and regular evaluations will assess the effectiveness of the interventions, allowing for iterative improvements. This approach ensures that the data collected are not only used to inform actionable outcomes but also lead to sustainable changes that enhance the work environment and well-being of ECNs.

Because we are following an action research process, management may choose to take actions based on this information as soon as practicable, and do not have to wait for the annual survey. However, their reactions and interventions must be recorded as part of the action research process. Overall, the main output from the surveys and focus groups will be information for the co-design meeting about what is working well for ECNs and what key things ECNs would advise their leaders/managers/organisation to improve. We know that challenges such as leadership inaction, resistance to change, and insufficient resources may arise. Addressing these challenges will involve securing commitment from senior leaders, implementing comprehensive change management strategies, and efficiently allocating resources. Further, differences in resources, leadership commitment, and healthcare systems can, and probably will, significantly impact this study’s implementation and scalability. Tailoring interventions to fit the specific context of each site and providing additional support where needed will be vital to aid in addressing these disparities.

#### 2.4.4. The Five-Year Cycle

The five-year cycle of the programme is achieved when each year a new group of ECNs is invited to participate, with each cohort advancing annually to ensure continuous development and insights are gained. For example, the first group of ECNs commenced in 2024 in New Zealand and will commence in 2025 at other sites with additional first-year ECNs in 2025/2026, and so on, up to fourth-year ECNs by 2028/2029 (Table 1).

It must be noted this progression and approach are ideal; however, it is important to recognise nuanced differences may occur across the various international study sites, both in terms of timing but also based on local needs and circumstances. Factors such as local practices and cultures, healthcare infrastructure, regulatory or ethical requirements, and resource availability may influence the implementation of the programme. Also, to mitigate the risk of participant disengagement over the five-year period, strategies may include providing nominal incentives (i.e., coffee vouchers), sending regular reminders, updates, and offering flexible participation options will be employed. These measures seek to maintain high levels of engagement and commitment among participants.

#### 2.4.5. Local Adaptation and Flexibility

Local adaptations will be managed through a flexible and context-sensitive approach that respects the different healthcare systems and cultural contexts of the participating sites. Each international study site will conduct an initial assessment to understand the local healthcare infrastructure, cultural norms, and specific needs of ECNs. This environmental scan will inform necessary adaptations to the research design and implementation. Engaging local stakeholders, including healthcare providers, policymakers, community leaders, and ECNs, will ensure that the adaptations are relevant and acceptable. The research process will incorporate iterative feedback loops, allowing for continuous refinement of interventions based on real-time feedback from participants. Collaborative decision-making within the international team will involve principal investigators, local research teams, and stakeholders to ensure adaptations are well-informed and contextually appropriate.

Examples of potential adaptations include adjustments in survey timing to align with local holidays, work schedules, or significant cultural events. For instance, surveys may be scheduled before or after extended public holidays to ensure higher response rates. Focus group formats may be adapted to suit local or health service preferences and logistical constraints, such as opting for smaller, more intimate groups or in-person sessions in regions with technological limitations. Surveys and focus group materials will be translated into local languages, such as Spanish. Local research teams will receive cultural sensitivity training to effectively engage with participants from diverse backgrounds. Interventions developed through the co-design process will be tailored to fit the specific cultural and organisational contexts of each site, such as different strategies for improving work–life balance based on local work culture and family dynamics. By managing local adaptations through these strategies, the research programme seeks to ensure interventions are both culturally adaptable, inclusive, and relevant to the diverse needs of ECNs and the health services where they are employed, thus enhancing their effectiveness and sustainability.

These variations require a flexible approach, allowing for adaptations that address the unique challenges and opportunities present in each location. By incorporating local insights and feedback, the programme can remain responsive and effective, ensuring it meets and addresses the various specific needs while maintaining overall consistency and quality.

## 3. Discussion

Prior to the COVID-19 pandemic, there was a global nursing workforce shortfall of around 5.9 million nurses. A significant historical challenge is the retention of ECNs, who often leave the profession due to stress, inadequate support, lack of professional development, and poor work–life balance [42]. The COVID-19 pandemic has exacerbated these issues, leading to widespread dissatisfaction [43]. Addressing inadequate organisational support and poor workplace cultures demands the creation of positive work environments that offer flexibility, adequate staffing, and supportive leadership. However, current practices often fall short in these areas, with many healthcare institutions struggling to provide the necessary support and resources [44].

Organisational support plays a crucial role in enhancing nurse well-being and retention. Zheng et al. [44] found that perceived organisational support directly impacts nurses’ occupational well-being and indirectly affects it through professional quality of life and the perception of decent work. This highlights the need for healthcare institutions to prioritise support measures to improve nurses’ well-being [45]. In contrast, many employers lack comprehensive support systems, leading to high turnover rates and dissatisfaction among nurses. Creating healthy work environments can significantly improve nurse retention and patient outcomes, yet many organisations still fail to implement robust systems to ensure the well-being and retention of their nursing staff [46].

Positive organisational culture is a critical factor in reducing work-related stress and promoting nurse retention. Kiptulon et al. [47] highlighted the impact of organisational culture on nurses’ stress levels, finding that positive organisational culture and climate are associated with lower levels of work-related stress. This aligns with the Improving Joy in Work framework, that emphasises the importance of understanding what matters to employees and addressing impediments to their well-being. However, many healthcare organisations are resistant to and fail to prioritise open dialogue and mutual respect between ECNs and nurse leaders, resulting in a less inclusive and responsive work environment [47].

The Thriving at Work model provides a pragmatic theoretical framework for understanding how organisational support might enhance nurse well-being by promoting a sense of vitality and continuous learning. This model proposes that both organisations and individuals are responsible for thriving at work, and that access to resources within organisations is crucial [48]. Current practices often overlook implementing approaches that promote vitality and continuous learning, leading to disengagement and higher turnover rates among nurses. There are a number of strategies presented in the literature advising managers how they can increase the well-being of the nursing workforce [49]. However, few of these are built on asking the people of concern, in this case ECNs, what matters to them. This first step is of major importance as attitudes, needs, and expectations are dynamic and may change as generational shifts change what is acceptable in and wanted from work [50]. This research programme engages with organisational nurse leaders/managers and then leverages their organisational commitment to engage in a co-design action research approach to help create positive organisational cultures and create work environments where nurses can thrive, ultimately benefiting both individual nurses and the overall effectiveness and sustainability of healthcare systems [51,52,53].

However, it must be recognised these organisational strategies are couched within broader systemic factors such as healthcare funding, staffing ratios, regulatory and policy environments, socioeconomic conditions, and technological advancements that also significantly influence organisational culture [54]. Insufficient funding and inadequate staffing and skill mix ratios can lead to resource constraints, increased workload, and burnout, while supportive policies and adequate resources can enhance nurse well-being and job satisfaction. Addressing these systemic challenges, although outside the remit of the programme of research, is also essential for creating a positive work environment where nurses can thrive.

Building on this theoretical platform by establishing an annual repeating cycle provides a systematic process that enables ECNs and senior nurse leaders to listen to each other and find mutually agreed improvements to the ways in which ECNs are supported and managed. This approach will enable innovation through the identification of local solutions. It will also provide a collaborative monitoring loop, as each year ECNs and nurse leaders will be able to re-examine what was said and decided the previous year, how those decisions were implemented, and what ensued. The results will be shared across the five study sites in different countries, so that new ideas can be shared and learnings rapidly mobilised across locations and where appropriate adapted across health service provider partners.

Lastly, this research programme is designed with a strong emphasis on cultural adaptability and relevance to ECNs. By incorporating culturally sensitive survey instruments and ensuring diverse representation in focus groups, the programme respects and acknowledges the varied cultural backgrounds of participating nurses. The flexibility of the programme allows for tailored interventions that align with the specific cultural norms and practices of different hospitals and countries. Preliminary studies and feedback from ECNs have highlighted the programme’s focus on critical areas such as burnout, organisational support, and leadership, demonstrating its direct relevance to the unique challenges faced by ECNs [54]. Furthermore, the co-design process actively involves ECNs and other stakeholders, ensuring that the interventions are not only culturally adaptable but also directly address the participants’ needs.

Overall, the proposed research approach offers a comprehensive approach to addressing the challenges faced by ECNs. By leveraging organisational support, co-design methodologies can promote positive organisational cultures where nurses can thrive and feel joy in work. This approach not only benefits individual nurses but may also contributes to the overall effectiveness and sustainability of healthcare systems. The integration of these elements is crucial for developing effective strategies to support and retain ECNs, ultimately leading to improved healthcare outcomes and a more resilient nursing workforce. However, the gap between current practices and these ideal strategies remains significant, highlighting the urgent need for healthcare organisations to adopt more effective and supportive measures [43].

## 4. Limitations

Several key limitations are noted within the proposed study and centre on the cultural and occupational variability across the five countries involved, which may affect the generalisability of the findings, as different healthcare systems and cultural contexts may influence outcomes. Additional limitations must be considered, including participant attrition, cultural variability, and the complexity of maintaining consistency of the research design across sites. Mitigating these limitations requires robust local protocols, reporting amendments, and regular communication across the international collaborative team while adapting interventions to fit the cultural context of each site. In addition, the longitudinal five-year duration of this study will require sustained engagement from participants and organisations and may be challenging to maintain. Operational variations, a consequence of each site perhaps executing the annual cycle and co-design process differently, may lead to difficulties comparing findings.

While co-design and co-production are valuable approaches, we cannot assume they will automatically lead to positive outcomes. Continuous monitoring and evaluation will be necessary to ensure these processes achieve the desired results. Lastly, it is noted there may be potential barriers, such as organisational resistance and resource limitations, that will need to be acknowledged. Addressing these barriers will require clear communication, demonstrating the benefits of interventions, and securing adequate funding. Overall, these factors demonstrate the complexity of conducting an extensive multi-centre international longitudinal research programme.

## Figures and Tables

**Figure 1 nursrep-15-00095-f001:**
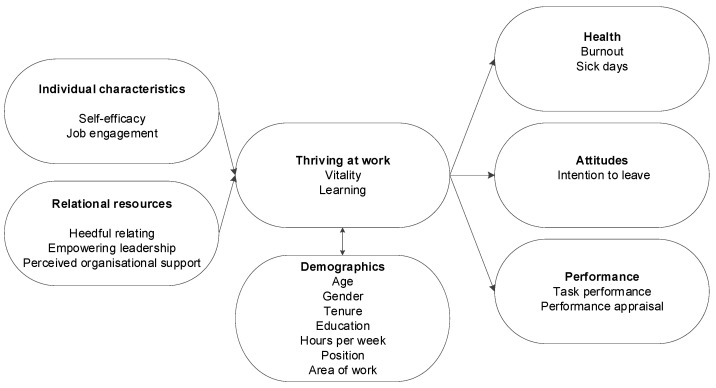
Revised conceptual model and network of assumed antecedents and outcomes of Thriving at Work, adapted from Kleine et al. [25].

**Figure 2 nursrep-15-00095-f002:**
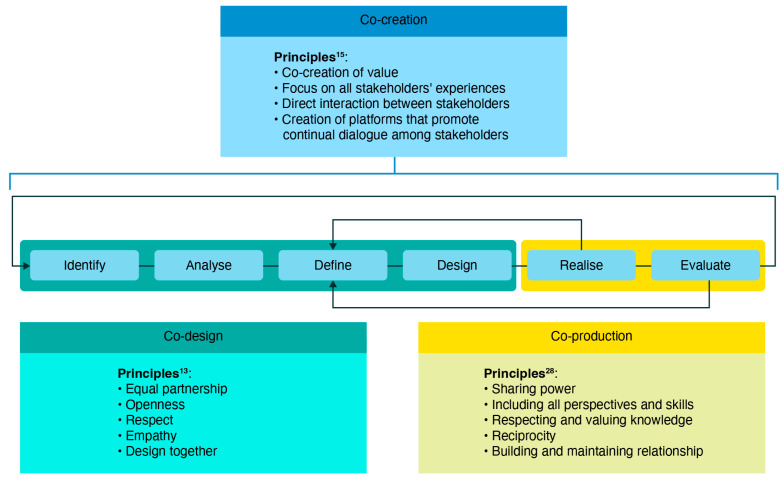
Co-creation model from Vargas et al. [40].

**Table 1 nursrep-15-00095-t001:** Five-year progression plan.

Group	2024/25	2025/26	2026/27	2027/28	2028/29
New Nurses	✔	✔	✔	✔	✔
End-of-First-Year ECNs		✔	✔	✔	✔
End-of-Second-Year ECNs			✔	✔	✔
End-of-Third-Year ECNs				✔	✔
End-of-Fourth-Year ECNs					✔

## Data Availability

No data were presented in this publication. In the future, data may be made available on request through the corresponding author.

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
