# Peer review of "Creating Organisational Working Conditions Where Nurses Can Thrive: An International Action Research Study"

_nursrep, 2025, doi:10.3390/nursrep15030095_

Round 1
Reviewer 1 Report
Comments and Suggestions for Authors
This is an excellent way to address the current issues facing healthcare systems today. The approach to this proposal is unique in that it provides an environment where nurse managers and frontline nurses may work together to discover solutions to enhance the work environment. It will be beneficial to compare the different sites to enhance learning of the best interventions that have been tested to make a difference with measurable outcomes.
Author Response
Reviewer #1
This is an excellent way to address the current issues facing healthcare systems today. The approach to this proposal is unique in that it provides an environment where nurse managers and frontline nurses may work together to discover solutions to enhance the work environment. It will be beneficial to compare the different sites to enhance learning of the best interventions that have been tested to make a difference with measurable outcomes.
Thank you for your review and positive comments of support. We have added a sentence to ensure the benefits of the multiple site research collaboration are made clear.
Reviewer 2 Report
Comments and Suggestions for Authors
Dear Authors,
Thank you for sharing your study protocol. Below few comments you might consider\clarify
The manuscript sometimes overgeneralizes or makes assumptions about universal applicability without adequate justification.
Methodological details, particularly regarding survey instruments and co-design processes, require greater elaboration.
How does the co-design approach differ from traditional interventions?
Avoid Overgeneralizations: Claims such as "50% of nurses leave within two years of graduating" and "burnout is the primary reason for attrition" should be contextualized with region-specific data or caveats about variability.
Support Theoretical Model Choice: Justify why the "Joy in Work" and "Thriving at Work" frameworks are appropriate for ECNs and international contexts.
Expand on Systemic Factors: Discuss broader systemic challenges (e.g., healthcare funding, staffing ratios) that influence organisational culture.
Address Generational Diversity: Consider differences in attitudes and expectations between Millennials and Gen Z nurses, as these may impact the applicability of interventions.
Explain how data from surveys, focus groups, and co-design meetings will be integrated into actionable outcomes.
Specify the validated tools to measure constructs like burnout, thriving, and organisational support.
Justify their cultural adaptability and relevance to ECNs.
Elaborate on how local adaptations will be managed, given the diverse healthcare systems and cultural contexts.
Provide examples of potential adaptations (e.g., adjustments in survey timing or focus group formats).
Acknowledge the risk of participant disengagement over five years and propose strategies to maintain engagement (e.g., incentives, reminders, flexible participation options).
Discuss challenges such as leadership inaction, resistance to change, or insufficient resources to implement interventions.
Specify how the success of interventions will be evaluated (e.g., retention rates, changes in survey scores, qualitative feedback).
Provide details on how ECNs will be protected when offering feedback on their organisations.
Consider mechanisms for anonymous feedback in co-design meetings.
Avoid assuming that co-design and co-production will automatically lead to positive outcomes.
Acknowledge potential barriers, such as organisational resistance or resource limitations.
Discuss how differences in resources, leadership commitment, and healthcare systems might affect the study’s implementation and scalability.
Provide a more detailed discussion of limitations, such as participant attrition, cultural variability, and the complexity of maintaining consistency across sites.
Strengthen References:
Ensure all cited data and frameworks are from up-to-date, reliable sources. Highlight evidence supporting the "Joy in Work" and "Thriving at Work" frameworks in nursing contexts.
Author Response
Reviewer #2
Dear Authors,
Thank you for sharing your study protocol.
Thank you for this statement. For clarity, this paper is not presenting a protocol. It is a programme of research and the research approach. We have made it clear in this document that each country will get ethical approval for its own protocol that is consistent with the laws and regulations in its county, while still following the overall approach presented in this paper. We have added additional context to the paper to make it clear that it is a programme of research being implemented by the international action research team.
Below few comments you might consider\clarify
The manuscript sometimes overgeneralizes or makes assumptions about universal applicability without adequate justification.
Methodological details, particularly regarding survey instruments and co-design processes, require greater elaboration.
Thank you for the comment and suggestion. We have now included the survey instrument, with questions along with a detailed overview of each survey item in the Appendix (Appendix A). In terms of the co-design process, this has been clearly outlined in detail from 2.1 through to 2.4 and any further explanation would be redundant.
How does the co-design approach differ from traditional interventions?
We have discussed in detail that most approaches are management designed and driven. We also outlined that a co-design approach is often use in the development of patient services but has not been used to any extent with the health workforce. We have also provided additional context for a generalised audience.
Avoid Overgeneralizations: Claims such as "50% of nurses leave within two years of graduating" and "burnout is the primary reason for attrition" should be contextualized with region-specific data or caveats about variability.
We have added a new reference re retention and turnover of ECNs. It now reads: A recent study of over 700,000 nurses throughout the US found that approximately 25 percent of nurses leave within the first year of graduating [6]. In Canada, 18% to 30% of ECNs choose to leave the nursing profession in their first year, and from 37% to 57% leave in their second year (8).
Support Theoretical Model Choice: Justify why the "Joy in Work" and "Thriving at Work" frameworks are appropriate for ECNs and international contexts.
Thank you for this comment. We have added additional context and justification regarding why the joy of work and thriving at work frameworks are appropriate for ECNs within section 1.2 entitled ‘A new approach’.
Expand on Systemic Factors: Discuss broader systemic challenges (e.g., healthcare funding, staffing ratios) that influence organisational culture.
Thank you for this suggestion. We have included a statement regarding the broader systematic challenges that need to be also considered within the discussion without taking away the thrust of the message regarding the program of research.
Address Generational Diversity: Consider differences in attitudes and expectations between Millennials and Gen Z nurses, as these may impact the applicability of interventions.
Thank you. We have added addition context within section 1.2, entitled ‘A new approach’ to address this suggestion.
Explain how data from surveys, focus groups, and co-design meetings will be integrated into actionable outcomes.
Thank you. We have provided addition detail within section 2.4, entitled ‘The research process’ when discussing the co-design meetings.
Specify the validated tools to measure constructs like burnout, thriving, and organisational support.
This has been addressed within the appendix – see appendix A
Justify their cultural adaptability and relevance to ECNs.
Thank you for this comment. We have addressed this within the discussion section of the manuscript to ensure clarity for readers.
Elaborate on how local adaptations will be managed, given the diverse healthcare systems and cultural contexts.
Thank you for this suggestion. We have provided addition detail within section 2.4, entitled ‘Local adaptations and flexibility’
Provide examples of potential adaptations (e.g., adjustments in survey timing or focus group formats).
Thank you for this suggestion. We have provided addition detail within section 2.4, entitled ‘Local adaptations and flexibility’
Acknowledge the risk of participant disengagement over five years and propose strategies to maintain engagement (e.g., incentives, reminders, flexible participation options).
Thank you for this suggestion. We have included a statement regarding this within the research process section of the paper.
Discuss challenges such as leadership inaction, resistance to change, or insufficient resources to implement interventions.
Thank you for this suggestion. We have included a statement regarding this within the research process section of the paper.
Specify how the success of interventions will be evaluated (e.g., retention rates, changes in survey scores, qualitative feedback).
Thank you for this suggestion. We have included a statement regarding this within the research process section of the paper.
Provide details on how ECNs will be protected when offering feedback on their organisations.
Thank you for this suggestion. We have included a statement regarding this within the research process section of the paper.
Consider mechanisms for anonymous feedback in co-design meetings.
We have considered this and have indicated that local adaptations will be needed to address these nuances. It must be noted the whole premise of co-design is working together; however, local adaptations may be required to support ECNs in this case.
Avoid assuming that co-design and co-production will automatically lead to positive outcomes.
Thank you for this comment. We have included a statement regarding this within the limitations section of the paper
Acknowledge potential barriers, such as organisational resistance or resource limitations.
Thank you for this suggestion. We have included a statement regarding this within the limitations section of the paper.
Discuss how differences in resources, leadership commitment, and healthcare systems might affect the study’s implementation and scalability.
Thank you for this suggestion. We have included a statement regarding this within the limitations section of the paper
Provide a more detailed discussion of limitations, such as participant attrition, cultural variability, and the complexity of maintaining consistency across sites.
Thank you for this suggestion. We have revised and provided a more detailed discussion of limitations in the limitations section of the manuscript.
Strengthen References:
Ensure all cited data and frameworks are from up-to-date, reliable sources. Highlight evidence supporting the "Joy in Work" and "Thriving at Work" frameworks in nursing contexts.
Thank you for this comment. We have used the original sources as these are seminal, and we have also included more recent references associated with the joy of work and thriving at work in nursing context throughout the document also.
Reviewer 3 Report
Comments and Suggestions for Authors
I appreciate the opportunity to review this interesting manuscript, which presents a protocol for an international study.
Despite the overall quality of the manuscript and its relevance, I would like to provide some recommendations that, in my humble opinion, could enhance the methodological rigour of the study’s exposition, thereby improving its readability and appeal to potential readers.
I suggest revising and condensing the abstract. While adhering to the standard abstract structure, it is important to note that this manuscript does not report on a completed study. Therefore, sections such as "Results" do not seem appropriate.
I also recommend restructuring and reorganising the introduction into four key content dimensions:
- Contextualisation
- Literature review and theoretical framework
- Justification and relevance of the topic
- Purpose and objectives of the study
Line 155–161: Would these not be expected results? I leave this question for the authors’ consideration.
Line 163: It would be beneficial to specify the universities involved.
The chapter titled “The Research Approach” should, in fact, be labelled “Materials and Methods”, as it corresponds to this section. I recommend restructuring this chapter to follow a classical format, such as:
- Study design
- Setting
- Participants
- Variables
- Instrumentation
- Data analysis and processing
- Ethical procedures
Given that this is a protocol for an empirical study with an international scope, it would be important for the final instruments to be available and included as appendices, or at the very least, for a detailed description of the variables to be provided. Similarly, there is a lack of detail regarding how the focus groups will be conducted, how data will be collected, processed, and analysed. Since this is a protocol, methodological clarity and precision are imperative; otherwise, the need for a protocol itself becomes questionable.
The inclusion of a discussion section does not seem justified. What exactly is being discussed? This concern is reinforced throughout the chapter, as it merely presents additional theoretical content without offering further insight. It might be more beneficial for the authors to elaborate on the study's relevance and anticipated findings, as well as their significance.
Additionally, the manuscript lacks a conclusion or final considerations.
Author Response
Reviewer #3
I appreciate the opportunity to review this interesting manuscript, which presents a protocol for an international study.
Despite the overall quality of the manuscript and its relevance, I would like to provide some recommendations that, in my humble opinion, could enhance the methodological rigour of the study’s exposition, thereby improving its readability and appeal to potential readers.
Thank you, we hope that we have addressed your comments successfully.
I suggest revising and condensing the abstract. While adhering to the standard abstract structure, it is important to note that this manuscript does not report on a completed study. Therefore, sections such as "Results" do not seem appropriate.
We appreciate this observation. The abstract has been condensed and a focus put on ‘Expected outcomes’ rather then results.
I also recommend restructuring and reorganising the introduction into four key content dimensions:
Contextualisation, Literature review and theoretical framework, Justification and relevance of the topic, Purpose and objectives of the study
Thank you for this suggestion. The introduction has been defined into the following categories: Contextualisation, Literature review and theoretical framework, Relevance to the research programme and Objectives and aims.
Line 155–161: Would these not be expected results? I leave this question for the authors’ consideration.
Thank you, we have considered and made the various revision where relevant.
Line 163: It would be beneficial to specify the universities involved.
The universities have been added here.
The chapter titled “The Research Approach” should, in fact, be labelled “Materials and Methods”, as it corresponds to this section. I recommend restructuring this chapter to follow a classical format, such as: Study design, Setting, Participants, Variables, Instrumentation, Data analysis and processing, Ethical procedures
Thank you. As indicated within the manuscript this is not a protocol, but a programme of study, where each international team will develop their own protocol based on the programme of study. A research programme of study is a broader, long-term plan that encompasses multiple related research projects or studies. It often addresses a larger research question or theme over an extended period. Although helpful for protocols and some research programmes of studies, ultimately this depends on the specific requirements of the research programme, the preferences of your research team, and the audience. The current structure effectively communicates the necessary information and meets the needs of the stakeholders. However, based on all reviews feedback we have adjusted the overall manuscript to ensure those vital elements are included and transparent.
Given that this is a protocol for an empirical study with an international scope, it would be important for the final instruments to be available and included as appendices, or at the very least, for a detailed description of the variables to be provided. Similarly, there is a lack of detail regarding how the focus groups will be conducted, how data will be collected, processed, and analysed. Since this is a protocol, methodological clarity and precision are imperative; otherwise, the need for a protocol itself becomes questionable.
Thank you. As indicated within the manuscript this is not a protocol, but a programme of study, where each international team will develop their own protocols. Nevertheless, we have provided additional insights and context to the manuscript including instrument in the appendix.
The inclusion of a discussion section does not seem justified. What exactly is being discussed? This concern is reinforced throughout the chapter, as it merely presents additional theoretical content without offering further insight. It might be more beneficial for the authors to elaborate on the study's relevance and anticipated findings, as well as their significance.
Thank you for your feedback. We understand your concern regarding the inclusion of the discussion section. The discussion is intended to highlight the study's relevance, anticipated findings, and their significance. Our research programme addresses critical challenges in nurse retention and well-being by applying the "Joy in Work" and "Thriving at Work" frameworks. These frameworks are designed to create supportive work environments that enhance job satisfaction and reduce burnout among ECNs. We anticipate that our findings will show improved nurse retention rates, increased job satisfaction, and better organizational support, which are crucial for both individual nurses and healthcare systems. By elaborating on these aspects, the discussion section provides valuable insights into the practical implications and broader impact of our study, thereby justifying its inclusion.
Additionally, the manuscript lacks a conclusion or final considerations.
Thank you for your feedback. We understand the importance of a conclusion or final considerations in a traditional research manuscript. However, given that this is a research programme of study, we believe that a formal conclusion may not be necessary at this stage. The nature of our research programme involves ongoing, iterative processes and continuous engagement with participants and stakeholders over several years. As such, the study is designed to evolve and adapt based on the findings and feedback collected throughout its duration. Instead of a traditional conclusion, we have focused on outlining the study's relevance, anticipated findings, and their significance within the discussion section. This approach allows us to provide a comprehensive overview of the programme's goals and expected impact while maintaining the flexibility required for a long-term, multi-site research initiative. We hope this explanation clarifies our rationale for not including a formal conclusion at this point.
Round 2
Reviewer 2 Report
Comments and Suggestions for Authors
Thank you for addressing the comments
Reviewer 3 Report
Comments and Suggestions for Authors
I appreciate once again the opportunity to participate in the review process of this manuscript. Regarding my recommendations, the authors have made the suggested revisions where applicable. Similarly, in cases where changes were not implemented despite the recommendation, the authors have provided justification and clarification for their methodological choices, effectively addressing my concerns and enhancing the manuscript's clarity. For this reason, I believe that the manuscript is now in a suitable condition for acceptance and publication.